# Sensory experience during locomotion promotes recovery of function in adult visual cortex

Megumi Kaneko[1,2], Michael P Stryker[1,2]*

[1]Center for Integrative Neuroscience, University of California, San Francisco, San Francisco, United States; [2]Department of Physiology, University of California, San Francisco, San Francisco, United States

**Abstract** Recovery from sensory deprivation is slow and incomplete in adult visual cortex. In this study, we show that visual stimulation during locomotion, which increases the gain of visual responses in primary visual cortex, dramatically enhances recovery in the mouse. Excitatory neurons regained normal levels of response, while narrow-spiking (inhibitory) neurons remained less active. Visual stimulation or locomotion alone did not enhance recovery. Responses to the particular visual stimuli viewed by the animal during locomotion recovered, while those to another normally effective stimulus did not, suggesting that locomotion promotes the recovery only of the neural circuits that are activated concurrent with the locomotion. These findings may provide an avenue for improving recovery from amblyopia in humans.

*For correspondence: stryker@phy.ucsf.edu

**Competing interests:** The authors declare that no competing interests exist.

**Reviewing editor**: Sacha B Nelson, Brandeis University, United States

## Introduction

Depriving one eye of normal patterned vision during early life causes a loss of visual acuity (amblyopia) and neurons in the visual cortex lose responsiveness through the deprived eye. If such abnormal vision is left uncorrected, recovery of visual function in adulthood is slow and incomplete, both in humans and in higher mammals, reflecting the limited plasticity of mature cortex (*Hubel and Wiesel, 1970*; *Mitchell and Sengpiel, 2009*). The recent discovery that neurons in mouse visual cortex respond with more than two times the number of action potentials during active locomotion than when still (*Niell and Stryker, 2010*) led us to test whether this enhanced response might facilitate recovery from the effects of early-onset, long-term monocular visual deprivation (MD).

## Results

Monocular deprivation was initiated by suturing shut the right eyelid of C57BL/6 mice early in the critical period (P22–24) and was continued to 4 to 5 months of age. The right eye was then re-opened to allow for binocular vision (BV), and baseline cortical responses through the two eyes were recorded using intrinsic signal imaging. Both eyes remained open afterward, and changes in responsiveness were measured over the next 3 weeks (*Figure 1A*). For 4 hr each day during these 21 days of BV (21d-BV), experimental animals viewed a visual stimulus (VS) while being permitted to run on a freely rotating spherical treadmill with their heads fixed. We used contrast-modulated stochastic noise matched to the spatiotemporal frequency response of the mouse as the visual stimulus because it drives nearly all cells in the primary visual cortex to some extent (*Niell and Stryker, 2008*).

In control mice that were kept in the standard housing condition (*home-cage*), cortical responses to the closed eye in binocular visual cortex slowly increased over 21d-BV, at which point they were still well below the range of age-matched mice with normal visual experience (*Figure 1B,D*). In contrast, mice that experienced the noise stimulus during their daily running (*VS+run*) showed a remarkable increase in closed-eye responses after only 7 days of BV (*Figure 1C,D*). Examined in a separate set of

**eLife digest** Amblyopia, otherwise known as 'lazy eye', is a condition in which vision fails to develop normally during childhood, not due to problems with the eye itself but due to problems with the transmission of information from the eye to the brain. It occurs when disorders such as squint—in which the eyes point in different directions—cause the brain to continually ignore input from one eye, with the result that vision in that eye never fully develops.

If detected in infancy, amblyopia can be treated by surgery, although such interventions must be performed early because they are much less effective when used on adults. However, Kaneko and Stryker now present data suggesting that the adult mammalian visual system may be more amenable to change than previously thought.

Young mice were deprived of visual input to one eye, by having an eyelid sewn shut, during a critical period in the development of their visual systems. When the eye was re-opened at the age of 4–5 months, the mice showed reduced responses in the brain region corresponding to that eye. However, if the mice were then allowed to run on a treadmill for several hours a day while viewing a visual stimulus—either black and white gratings or random noise—the vision in their deprived eye showed a rapid and striking improvement. This improvement was not seen in mice that ran without a visual stimulus, or in mice that looked at the visual stimulus but did not run. Moreover, the improvement was specific to the particular stimulus viewed whilst running.

Although the mechanism behind this effect is unclear, it is known that running increases neuronal activity, and one possibility is that neurons that are active simultaneously—such as those encoding the visual stimulus—form stronger connections with one another: 'neurons that fire together, wire together'. Further work is required to determine whether similar changes occur in the human visual system and, if they do, whether they could be applied to the treatment of amblyopia.

animals, this effect of *VS+run* was already significant after 3 days of training (*Figure 1—figure supplement 1*). Responses through the open eye did not change significantly in either group (*Figure 1B,C,E*). As a result, the ocular dominance index (ODI), computed as a normalized difference between responses to two eyes, recovered much more rapidly in animals with visual experience during running than in controls (*Figure 1F*). Recovery using reverse occlusion (switching the eye closure) instead of BV was similarly enhanced by *VS+run* (*Figure 1—figure supplement 2*).

To determine whether VS or running alone enhances recovery, we tested two more groups of mice: one that ran on the ball without VS (*run-only*) and one that viewed the visual noise stimulus in the home cage but did not run on the ball (*VS-only*). Closed-eye responses and ODI in both of these groups increased only slowly, similar to those of the *home-cage* control group (*Figure 1G,I*). Running velocity and duration were similar between *run-only* and *VS+run* groups (*Figure 1—figure supplement 3*).

*VS+run* enhanced recovery of monocular visual areas as well (*Figure 2*). This effect was rapid (*Figure 2—figure supplement 1*) and was particularly prominent for a secondary monocular area, where the *home cage* and the *run-only* groups showed no significant improvement even after 21 days of BV.

Perception in humans can be strengthened by training in an experience-specific manner (reviewed in *Sagi, 2011*), and responses in mouse visual cortex are reported to do so as well (*Frenkel et al., 2006*). We next tested whether recovery of closed-eye responses is preferentially enhanced to the particular visual stimuli presented during running. Mice viewed either square bars drifting in eight evenly-spaced directions (*barVS*) or the contrast-modulated noise (*noiseVS*) during running each day for 3 weeks. Intrinsic signal responses to both bar and noise stimuli were measured weekly. Mice that experienced *barVS+run* showed significantly greater recovery of closed-eye responses to bars than to noise (*Figure 3*). Likewise, mice that experienced *noiseVS+run* showed significantly greater recovery of response to noise than to bars (*Figure 3*). Responses to the stimulus that was not experienced during running were similar to those in control mice presented with a blank screen during running. These observations reveal that enhanced recovery is stimulus-specific and suggest that only the specific visual cortical circuits that are active during locomotion recover.

To reveal changes in response properties of individual neurons, we made extracellular single unit recordings from neurons in layers 2/3 and 4 of binocular V1 contralateral to the closed eye. The use of

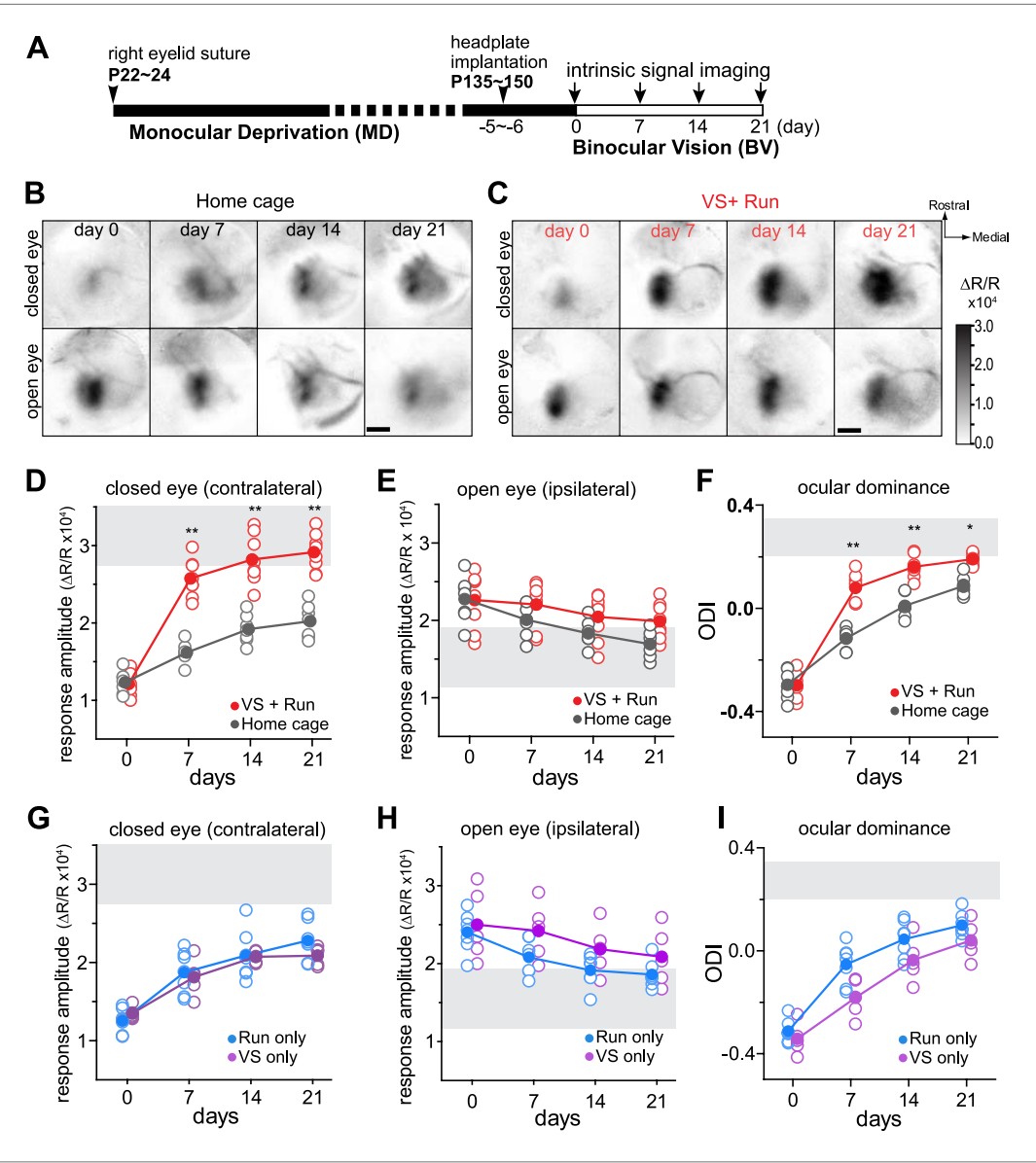

**Figure 1**. Visual stimulation during locomotion enhances recovery of cortical responses through the deprived eye after prolonged MD. (**A**) Experimental schedule to examine changes in visual cortical responses over 21d-BV following prolonged MD started at postnatal day (P) 22–24. (**B** and **C**) Examples of intrinsic signal responses to the closed eye in the binocular visual cortex during 21d-BV in a *home-cage* control mouse (**B**) and in a mouse viewing contrast-modulated noise as VS during daily runs (*VS+run*, **C**). (**D** and **E**) Changes in intrinsic signal responses evoked by the noise through the closed (**D**) and open (**E**) eyes in *home-cage* (n = 8) and *VS+run* mice (n = 8). (**F**) Ocular dominance index (ODI) computed from response amplitude to contralateral (closed) and ipsilateral (open) eyes shown in **D** and **E**. ODI represents normalized difference in response magnitude between two eyes with 0 being equal amplitude to two eyes; the higher the number, more contralateral eye dominant. (**G** and **H**) Changes in intrinsic signal responses evoked by the noise through the closed (**G**) and open (**H**) eyes in *run-only* (n = 7) and *VS-only* mice (n = 7). (**I**) Ocular dominance index (ODI) computed from response amplitude to contralateral (closed) and ipsilateral (open) eyes shown in **G** and **H**. Gray area in **D**–**I** indicates the range of response amplitude or ODI in age-matched mice with normal visual experience. **p<0.01, *p<0.05, between groups.

The following figure supplements are available for figure 1:

**Figure supplement 1**. Effects of visual stimulation during locomotion after 4 days of binocular vision.

*Figure 1. Continued on next page*

*Figure 1. Continued*

**Figure supplement 2**. Changes in responses during reverse occlusion measured using intrinsic signal imaging.

**Figure supplement 3**. Monitoring of locomotion while the mice are on the track-ball.

16-site silicon probes and spike-sorting methods (*Niell and Stryker, 2008*) enabled unbiased sampling from both responsive and unresponsive units. We examined five groups of mice with different treatments (*Figure 4A*). We first describe results from the broad-spiking, presumed excitatory cells that constitute ~80% of the recordings.

Consistent with the intrinsic signal observations as described above, closed-eye responses in mice that viewed visual stimuli while running during 7d-BV recovered significantly from their profoundly decreased level after long-term monocular deprivation (LTMD), in a stimulus-specific manner (*Figure 4*, *Figure 4—figure supplements 1, 2*). Closed-eye responses to the optimal grating (preferred orientation and spatial frequency) were most increased in *grating+run* mice among groups (*Figure 4B*), while responses to contrast-modulated noise were improved most in *noise+run* mice (*Figure 4C*). Open-eye responses did not differ among four groups of monocularly deprived mice (*Figure 4—figure supplement 1*). As a result, ocular dominance recovered to normal levels with similar stimulus specificity (*Figure 4B,C*).

In normal animals, the majority of cells in layers 2–4 are orientation-selective (OSI >0.5, *Figure 5A,B*) and the OSI for the two eyes is similar in each binocular cell (*Figure 5—figure supplement 1B*). Closed-eye OSIs and the binocular correlation of OSI were significantly improved only in *grating+run* mice but not in other groups (*Figure 5A,B*, *Figure 5—figure supplement 1B*). Prolonged MD caused mismatch of the preferred orientation of two eyes (*Figure 5C,D*), consistent with requirement of binocular visual experience during the critical period to develop orientation matching (*Wang et al., 2010*). 7d-BV improved binocular matching almost to normal levels in *grating+run* but not in *home-cage* or *noise+run*, mice (*Figure 5C,D*). In addition, the distribution of the preferred spatial frequency in closed-eye responses was almost normalized in *grating+run* mice but not in *home-cage* or *noise+run* mice. (*Figure 5E,F*). Such shifts in preferred spatial frequency may account for improvements in behaviorally measured acuity (*Mitchell and Sengpiel, 2009*).

In response to the contrast-modulated noise stimuli, the peak firing of a cell occurs at the frequency of contrast modulation (F1, at 0.1 Hz). We assessed contrast sensitivity by calculating average value of

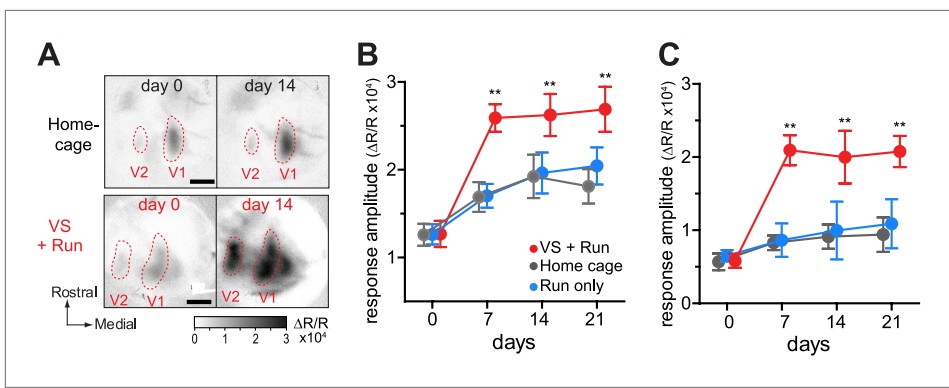

**Figure 2**. Visual stimulation during locomotion enhances recovery of cortical responses in the monocular visual cortex after prolonged MD. (**A**) Examples of intrinsic signal images of monocular visual areas. (**B** and **C**) Changes in intrinsic signal magnitudes (mean ± SEM) through the closed eye in response to the noise in the monocular V1 (**B**) and monocular secondary visual cortex (**C**) (same animals shown in *Figure 1D–F*). **p<0.01, between groups.

The following figure supplements are available for figure 2:

**Figure supplement 1**. Effects of visual stimulation during locomotion on responses in the monocular zone after 4 days of binocular vision.

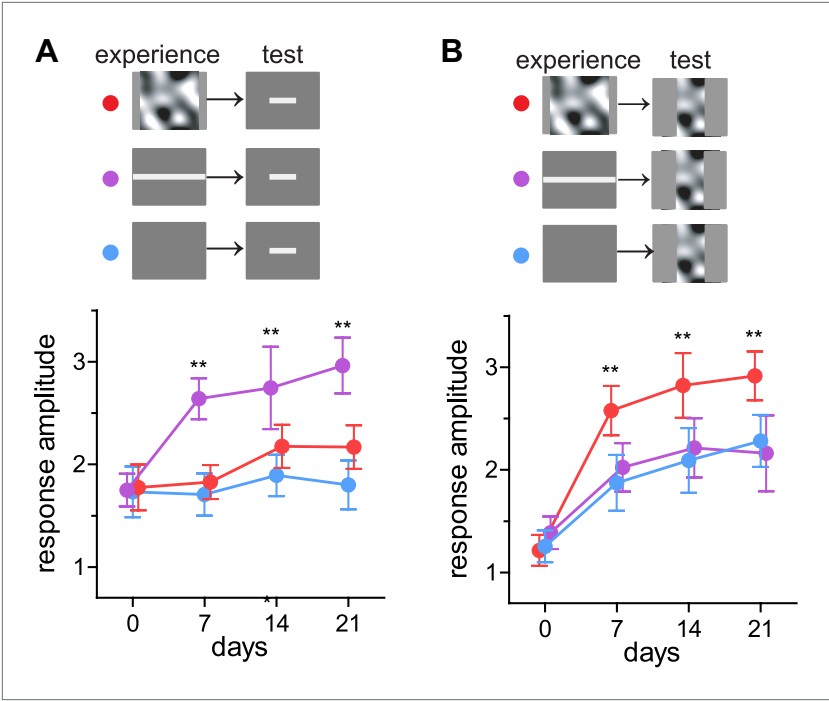

**Figure 3**. Preferential enhancement of recovery of closed-eye responses to the visual stimuli experienced during locomotion. (**A**) Peak intrinsic signal amplitude in response to bar stimuli through the closed eye in mice that experienced noise (*noiseVS+run*, red, n = 6), drifting bars (*barVS+run*, n = 6), or a blank screen during running (blue, n = 6). (**B**) Peak intrinsic signal amplitude in response to bar stimuli in same mice as in **A**. Data are show mean ± SD. **p<0.01 and *p<0.05 compared with the blank-screen control (blue).

contrast that elicits half-maximal response ($C_{1/2}$). Contrast sensitivity was significantly impaired after LTMD and was restored almost completely in *noise+run* mice but not in *gratings+run* mice (*Figure 6*).

Narrow-spiking cells are thought to correspond to inhibitory, predominantly fast-spiking, interneurons (*McCormick et al., 1985*; *Bartho et al., 2004*). Although a minority population, inhibitory cells play an important role in plasticity (*Espinosa and Stryker, 2012*). We observed several notable differences between broad- and narrow-spiking cells in the changes following LTMD and 7d-BV. First, spontaneous activity of narrow-spiking cells was greatly reduced after LTMD and did not change significantly after 7d-BV regardless of the treatment (*Figure 7C,D*), whereas those of broad-spiking cells were elevated after LTMD and significantly decreased toward normal level in mice after 7d-BV (*Figure 7A,B*). These opposite changes in spontaneous firing between broad- and narrow-spiking cells may reflect a homeostatic mechanism that alters excitatory–inhibitory balance to maintain cortical activity during prolonged deprivation. Second, recovery of closed-eye responses and ocular dominance to the gratings in narrow-spiking cells was only modest even in *grating+run* mice (*Figure 8A*, *Figure 8—figure supplement 1A*), whereas it was nearly complete in broad-spiking cells (*Figure 4B*). Third, responses of narrow-spiking cells to contrast-modulated noise through the deprived eye and ocular dominance were incompletely restored both in *noise+run* and *grating+run* mice to similar extents, showing no preference for the experienced stimulus (*Figure 8B*, *Figure 8—figure supplement 1B*). Fourth, responses of narrow-spiking cells through the open-eye were not elevated after LTMD either to the noise or grating stimulus, and did not change significantly in any of 7d-BV mice (*Figure 8—figure supplement 1A,B*). This lack of potentiation of open-eye responses during LTMD in narrow-spiking cells may contribute to potentiation of those in broad-spiking cells. The responses of narrow-spiking cells after LTMD suggest that deprivation reduces intracortical inhibition. Such a reduction may provide a starting point that allows meager deprived-eye excitatory pathways of the cortical circuit to drive activity upon re-opening. The incomplete recovery of orientation tuning of broad-spiking cells in spite of nearly full restoration of the magnitudes of responses may result from the very limited recovery in narrow-spiking

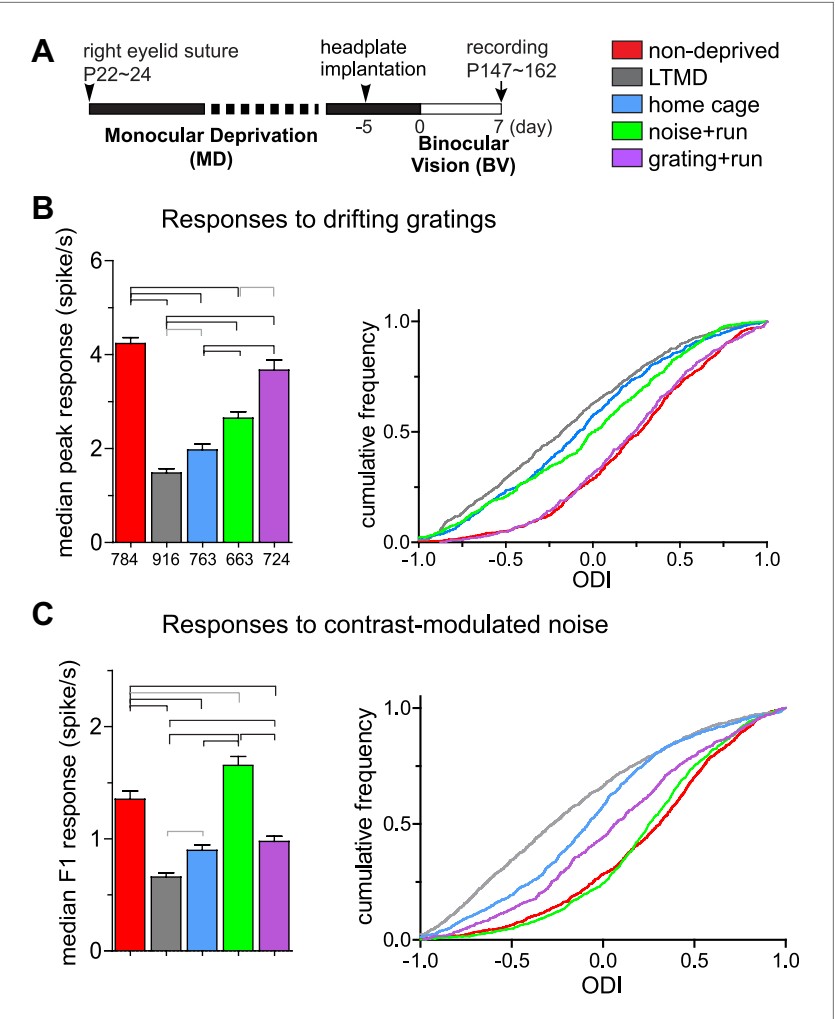

**Figure 4**. Response magnitude of individual broad-spiking cells to drifting gratings and contrast-modulated noise. (**A**) Experimental schedule for single unit recording. Color-coded bars for different treatment groups apply to all panels. (**B**) Response magnitude to drifting gratings. Left panel: median (±s.e.) response rates to optimal drifting gratings through the deprived eye. Right panel: cumulative frequency distribution of ocular dominance index (ODI) in cells that were responsive (>2 spikes/s) through either deprived or open eye. ODI was computed from response magnitude to optimal gratings through each eye as shown in *Figure 4—figure supplement 1A*. (**C**) Response magnitude to contrast-modulated noise. Left: median F1 response (±s.e.) to the noise stimulus through the deprived eye. Right: cumulative frequency distribution of ODI calculated for each cell that were responsive to the noise (F1 response >0.2) through either eye from the data shown in *Figure 4—figure supplement 1B*. Data in left panels of **B** and **C** are from same populations of all cells isolated, as numbers are indicated below bars in **B**. Horizontal lines above bars; black: p<0.01, gray: p<0.05. Results of Kolmogorov–Smirnov tests for cumulative frequency distributions are shown in *Table 1*.

The following source data, source code and figure supplements are available for figure 4:

**Figure supplement 2—Source data 1**. This refers to Panel A. Responses of 916 broad-spiking cells to two different stimuli in spikes/sec in LTMD animals without subsequent binocular vision. Column 1 shows peak responses to the optimal grating. Column 2 shows response to the contrast modulated noise pattern at the fundamental (F1) frequency of contrast modulation.

**Figure supplement 2—Source data 2**. This refers to Panel B. Responses of 663 broad-spiking cells to two different stimuli in spikes/sec in *run+noise* mice. Columns as in *Figure 4—Figure supplement 2—source data 1*.

**Figure supplement 2—Source data 3**. This refers to Panel C. Responses of 724 broad-spiking cells to two different stimuli in spikes/sec in *run+gratings* mice. Columns as in *Figure 4—Figure supplement 2—source data 1*.

*Figure 4. Continued on next page*

*Figure 4. Continued*

**Figure supplement 2—Source code 1**. Computer code using earth mover distance algorithm for analysis of fictive changes in response.

**Figure supplement 1**. Response magnitudes of individual broad-spiking cells to drifting gratings and contrast-modulated noise.

**Figure supplement 2**. Analysis of fictive longitudinal data on the stimulus specificity of recovery during locomotion.

cells, as inhibition is critical for generation of sharp orientation tuning in upper layer neurons in the primary visual cortex (*Liu et al., 2011*).

## Discussion

Recovery in adulthood from amblyopia induced by monocular visual deprivation during early life is slow and far from complete, even weeks after opening the deprived eye (*Hubel and Wiesel, 1970*; *Mitchell and Sengpiel, 2009*). In this study, we have used intrinsic signal imaging and single cell recording to find that locomotion, which is known to increase visual responses (*Niell and Stryker, 2010*), dramatically enhances the recovery of visual responses, almost to normal levels within a week. Surprisingly, the recovery of response was specific to the particular visual stimuli presented during locomotion. Neither locomotion alone nor visual stimulation alone promoted recovery. These findings suggest that recovery is facilitated only in the neural circuits that are activated during running.

Other manipulations have been reported to enhance adult plasticity (*Bavelier et al., 2010*), such as dark exposure (*He et al., 2007*), environmental enrichment (*Sale et al., 2007*), and antidepressant treatment (*Maya Vetencourt et al., 2008*). These treatments have in common a decrease in intracortical inhibition, consistent with our finding of reduced activity of inhibitory neurons. Such treatments may also increase locomotion, producing an active cortical state that facilitates plasticity. Interestingly, active hunting by barn owls, flying rather than walking or running in this case, dramatically increases adaptive plasticity of the auditory map in the superior colliculus (*Bergan et al., 2005*).

A number of mechanisms may contribute to the enhanced recovery described here. Locomotion may induce activity in neuromodulatory circuits that might enhance cortical function or plasticity (*Carcea and Froemke, 2013*), and it may also produce hemodynamic changes that might affect the operation of the cortical circuit (*Seifert and Secher, 2011*). The stimulus specificity of enhanced recovery, however, rules out any general explanation of recovery as an increase in cortical response to any stimulus. Most simply, the larger responses of the cortical neurons themselves during locomotion would be expected to enhance recovery through the operation of any Hebbian or spike-timing based plasticity mechanism that depends on neural activity (*Feldman, 2009*). Plasticity of excitatory connections would be further enhanced by the opposite changes in spontaneous firing of broad- and narrow-spiking neurons.

The present findings do not allow a rigorous test of the hypothesis that Hebbian changes are responsible for the stimulus specificity of recovery; i.e., that the neurons or synapses that are driven better by the specific stimuli presented during locomotion become more powerful. Such a test would require monitoring of the responses of individual cells or the efficacy of the synapses that they make throughout the recovery period. However, responses to both grating and noise stimuli were measured in three large populations of broad-spiking neurons, one before recovery and two in separate groups of animals after recovery during exposure to grating or noise. Making a correspondence between the individual neurons in these samples, in order to create fictive repeated measures from the same neurons, may suggest whether the present data are compatible with the Hebbian hypothesis. Indeed, the fictive longitudinal data show that noise stimulation increased the responses of neurons originally responding better to noise more than it did for the neurons that responded better to grating, and vice versa (*Figure 4—figure supplement 2*). These findings are consistent with the Hebbian hypothesis.

In human visual cortex, it is unknown whether neuronal responses or plasticity are enhanced by locomotion. In monkeys, attention modulates the gain of cortical responses in higher visual areas, but these effects

are very much smaller in V1 and smaller than those of locomotion in mice (*Maunsell and Cook, 2002*; *Reynolds and Chelazzi, 2004*). It will be interesting to determine whether V1 responses in humans are enhanced by locomotion and whether recovery from amblyopia in humans can be similarly enhanced.

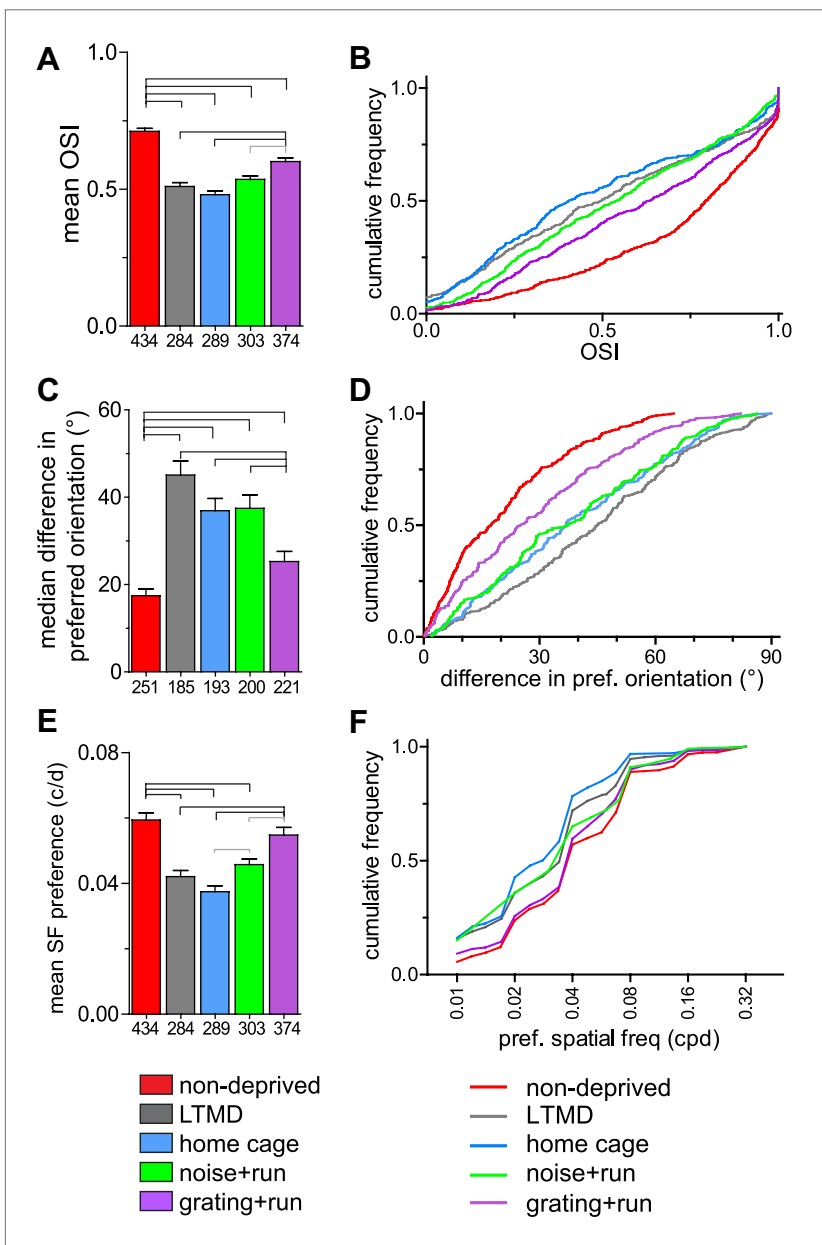

**Figure 5**. Tuning properties of individual broad-spiking cells in response to drifting gratings. (**A** and **B**) Orientation tuning of deprived-eye responses, expressed as mean (±SEM) orientation selectivity index (OSI) (**A**) and cumulative frequency distribution of OSI (**B**). (**C** and **D**) Binocular matching of preferred orientation. Absolute differences in preferred orientation between two eyes in binocularly responsive cells are presented as the median (±s.e.) (**C**) and the cumulative frequency distribution (**D**). (**E** and **F**) Spatial frequency tuning. Preferred spatial frequencies at the preferred orientation of drifting gratings through the deprived eye are shown as mean (±SEM) (**E**) and cumulative frequency distribution (**F**). Sample sizes are indicated below bars. Horizontal lines above bars; black: $p<0.01$, gray: $p<0.05$. Results of Kolmogorov–Smirnov tests for **B**, **D**, **F** are shown in *Table 1*.

The following figure supplements are available for figure 5:

**Figure supplement 1**. Tuning properties of broad-spiking cells examined with drifting sinusoidal gratings.

**Table 1.** Results of Kolmogorov-Smirnov tests for cumulative frequency distributions

| Groups compared | Non-deprived LTMD | Non-deprived Home cage | Non-deprived noise+run | Non-deprived grating+run | LTMD Home cage | LTMD noise+run | LTMD grating+run | Home cage noise+run | Home cage grating+run | noise + run grating+run |
|---|---|---|---|---|---|---|---|---|---|---|
| **Broad-spiking cells** | | | | | | | | | | |
| Spontaneous firing | <0.00001 | <0.0001 | <0.001 | <0.05 | <0.01 | <0.01 | <0.001 | <0.01 | <0.05 | <0.01 |
| Grating response: closed | <0.00001 | <0.00001 | <0.01 | n.s. | n.s. | <0.0001 | <0.05 | <0.05 | <0.0001 | <0.01 |
| Grating response: open | <0.01 | <0.05 | <0.05 | n.s. | n.s. | n.s. | n.s. | n.s. | n.s. | n.s. |
| Grating response: ODI | <0.00001 | <0.00001 | <0.0001 | n.s. | <0.01 | <0.001 | <0.00001 | <0.01 | <0.00001 | <0.001 |
| \|dO\| | <0.00001 | <0.00001 | <0.00001 | <0.01 | n.s. | n.s. | <0.001 | n.s. | <0.001 | <0.001 |
| OSI: closed | <0.00001 | <0.00001 | <0.00001 | <0.00001 | n.s. | <0.05 | <0.001 | <0.001 | <0.001 | <0.01 |
| OSI: open | <0.01 | <0.01 | <0.05 | <0.05 | n.s. | n.s. | n.s. | n.s. | n.s. | n.s. |
| Orientation tuning width: closed | <0.0001 | <0.05 | <0.05 | <0.05 | n.s. | n.s. | <0.05 | n.s. | n.s. | n.s. |
| Orientation tuning width: open | n.s. | n.s. | n.s. | n.s. | n.s. | n.s. | n.s. | n.s. | n.s. | n.s. |
| Preferred SF: closed | <0.001 | <0.00001 | <0.001 | n.s. | n.s. | n.s. | <0.05 | <0.05 | 0.001 | <0.05 |
| Preferred SF: open | <0.00001 | <0.001 | <0.001 | <0.001 | <0.01 | <0.01 | <0.01 | n.s. | n.s. | n.s. |
| Noise response (F1): closed | <0.00001 | <0.0001 | <0.01 | <0.001 | <0.05 | <0.00001 | <0.001 | <0.00001 | <0.05 | <0.0001 |
| Noise response (F1): open | <0.0001 | <0.0001 | <0.0001 | <0.00001 | n.s. | n.s. | n.s. | n.s. | n.s. | n.s. |
| Noise response: ODI | <0.00001 | <0.00001 | n.s. | <0.0001 | <0.0001 | <0.00001 | <0.0001 | <0.00001 | <0.001 | <0.001 |
| **Narrow-spiking cells** | | | | | | | | | | |
| Spontaneous firing | <0.00001 | <0.00001 | <0.00001 | <0.00001 | n.s. | n.s. | n.s. | n.s. | n.s. | n.s. |
| Grating response: closed | <0.00001 | <0.0001 | <0.0001 | <0.001 | n.s. | n.s. | <0.01 | n.s. | <0.05 | <0.05 |
| Grating response: open | n.s. | n.s. | n.s. | n.s. | n.s. | n.s. | n.s. | n.s. | n.s. | n.s. |
| Grating response: ODI | <0.00001 | <0.001 | <0.0001 | <0.05 | <0.01 | n.s. | <0.001 | <0.001 | n.s. | <0.05 |
| OSI: closed | <0.001 | <0.0001 | n.s. | n.s. | n.s. | n.s. | n.s. | <0.01 | <0.01 | n.s. |

*Table 1. Continued on next page*

*Table 1. Continued*

| Groups compared | Non-deprived | | | | LTMD | | | Home cage | | noise + run |
| --- | --- | --- | --- | --- | --- | --- | --- | --- | --- | --- |
| | LTMD | Home cage | noise+run | grating+run | Home cage | noise+run | grating+run | noise+run | grating+run | grating+run |
| OSI: open | <0.01 | <0.01 | <0.01 | <0.01 | n.s. | n.s. | n.s. | n.s. | n.s. | n.s. |
| Noise response (F1): closed | <0.001 | <0.001 | <0.01 | <0.01 | <0.05 | <0.01 | <0.05 | <0.01 | <0.01 | n.s. |
| Noise response (F1): open | n.s. | n.s. | n.s. | n.s. | n.s. | n.s. | n.s. | n.s. | n.s. | n.s. |
| Noise response: ODI | <0.00001 | <0.00001 | <0.001 | <0.01 | <0.001 | <0.0001 | <0.0001 | <0.001 | <0.001 | <0.05 |

Statistical significance levels are indicated after adjusting p-values using Bonferroni correction for multiple comparisons. n.s.:p>0.05. closed: measures through deprived eye, open: measures through open eye. ODI: ocular dominance index. |dO|: absolute difference in preferred orientation between left and right eyes. OSI: orientation selectivity index.

# Materials and methods

## Animals, monocular deprivation, and running on a spherical treadmill

C57BL/6 wild-type breeders were purchased from Jackson Laboratory (Bar Harbor, ME) and bred as needed. Animals were maintained in the animal facility at University of California San Francisco and used in accordance with Protocol AN098080-01D approved by the UCSF Institutional Animal Care and Use Committee. Monocular deprivation (MD) was performed as described (*Gordon and Stryker, 1996*) except that 2–3% isoflurane in oxygen was used for anesthesia. The lid of the right eye was sutured shut at P22–24. Mice were housed in the standard condition (12/12 dark–light cycle, free access to food and water) until P135~150, at which time a custom stainless steel plate for head fixation was attached to the skull with dental acrylic under isoflurane anesthesia. The exposed surface of the skull was covered with a thin coat of nitrocellulose (New-Skin, Medtech Products Inc., NY) to prevent desiccation, reactive cell growth, and destruction of the bone structure. Animals were given a subcutaneous injection of carprofen (5 mg/kg) as a post-operative analgesic. 5–7 days after head-plate implantation, the closed eyelid was re-opened and groups of mice for the intrinsic signal imaging study underwent the first imaging session (day 0). The re-opened eyelid was left open afterward to allow binocular vision while animals were subjected to the different regimes of visual stimulation and locomotion as follows. (1) *running with visual stimulation*: The mouse was allowed to freely move its limbs and trunk on a foam ball floated on a stream of air, while its head was fixed via the implanted headplate that could be screwed into a rigid crossbar above the floating ball as described (*Niell and Stryker, 2010*). Two optical mice were used to measure the displacement of the ball as the mouse moved, allowing us to calculate the physical speed of the ball (*Niell and Stryker, 2010*). After re-opening the deprived eye, each animal spent 4 hr daily on the ball while viewing visual stimulus, over next 3 weeks (for intrinsic signal optical imaging) or 6 days (for single unit recording). Running was performed during the dark phase of the housing cycle, because we found that mice ran for longer times during this phase. (2) *running without visual stimulation*: Each animal ran on the floating ball exactly as the first group but viewed only a blank, mid-gray screen (~35 lux). (3) *visual stimulation alone without running (intrinsic signal imaging only)*: a transparent cage containing a

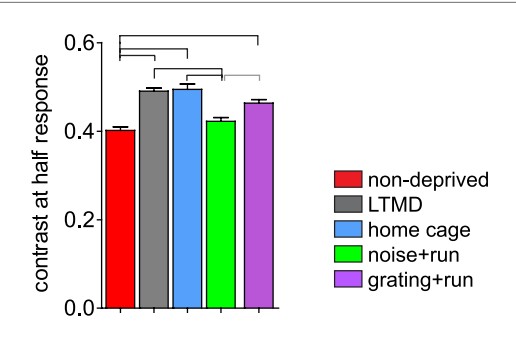

**Figure 6**. Change in contrast sensitivity in broad-spiking cells. Average values of contrast that gives half-maximal response are shown. Horizontal lines above bars; black: $p < 0.01$, grey: $p < 0.05$.

group of mice with regular bedding materials was surrounded by four monitors with the distance of 25 cm between the center of the cage and the monitor. The visual stimulus was presented on the monitors continuously for 7–8 hr daily during which animals were allowed to behave freely without disruption by the experimenter. We extended the duration to 7–8 hr because animals were obviously active only 50–60% of time. (4) *no running, no visual stimulation*: mice were housed in the standard condition without running or specific visual stimulation. (5) *Non-deprived control for single unit recording only*: age-matched control animals, many of them being in the same litter as those in experimental groups, were housed in the standard condition with normal visual experience without running or specific visual stimulation.

## Visual stimuli

The daily visual stimulus for animals used in intrinsic signal imaging was either contrast-modulated noise movies or drifting square bars at the full range of orientations. The stimulus for mice employed in single unit recordings was either the noise or drifting sinusoidal gratings. Stimuli were generated in Matlab using Psychophysics Toolbox extensions (*Brainard, 1997*; *Pelli, 1997*), and displayed on a LCD monitor (60-Hz refresh rate) placed 25 cm from the mouse, spanning 52.5° (height) × 70° (width) of visual space. The drifting bar was full-length of the monitor, width of 2°, velocity of 25°/s, and drifting in eight equally spaced directions. The drifting sinusoidal gratings were shown for a duration of 1.5 s, at temporal frequency of 2 Hz, spatial frequency of 0.01, 0.02, 0.04, 0.08, 0.16, 0.32, and 0 (full-field flicker) cycles/° (cpd), in 12 equally spaced directions. The contrast-modulated Gaussian noise movie consisted of the Fourier-inversion of a randomly generated spatiotemporal spectrum with low-pass spatial and temporal cutoffs applied at 0.05 cpd and 4 Hz, respectively. To provide contrast modulation, the movie was multiplied by a sinusoid with a 10-s period. Movies were generated at 60 ×60 pixels and then smoothly interpolated by the video card to 480 × 480 to appear 30 × 30 cm on the monitor and played at 30 frames per second. Each movie was 5 min long and repeated for 4 hr total presentation.

## Optical imaging of intrinsic signals

5–7 days after the headplate implantation, the first imaging of intrinsic signals was performed to measure baseline responses through each eye. The mouse was anesthetized with isoflurane (3% for induction and 0.7% during recording) supplemented with intramuscular injection of chlorprothixene chloride (2 µg/g body weight), and the closed eyelid was carefully opened by slitting horizontally at the center of the fused lid just before the imaging session. Repeated optical imaging of intrinsic signals was performed as described (*Kaneko et al., 2008*). We monitored the concentration of isoflurane using an Ohmeda 5250 RGM (Datex-Ohmeda, Madison, WI) throughout each imaging session. Images were recorded transcranially through the window of the implanted headplate. Intrinsic signal images were obtained with a Dalsa 1M30 CCD camera (Dalsa, Waterloo, Canada) with a 135 × 50 mm tandem lens (Nikon Inc., Melville, NY) and red interference filter (610 ± 10 nm). Frames were acquired at a rate of 30 fps, temporally binned by four frames, and stored as 512 × 512 pixel images after binning the 1024 × 1024 camera pixels by 2 × 2 pixels spatially. We used two kinds of visual stimuli presented between −5° and 15° (azimuth) on the stimulus monitor (0° = center of the monitor aligned to center of the mouse) to record intrinsic signals in the binocular visual cortex: (1) 2°-wide bars, moving continuously and periodically upward or downward at a speed of 10°/sec; (2) the contrast-modulated noise movie, as described above. To record in the monocular visual area, the contrast-modulated noise was presented between 50° and 70° (azimuth) of the visual field. Visual stimuli were presented on a 40 × 30 cm monitor placed 25 cm in front of the mouse. The phase and amplitude of cortical responses at the stimulus frequency were extracted by Fourier analysis as described (*Kalatsky and Stryker,*

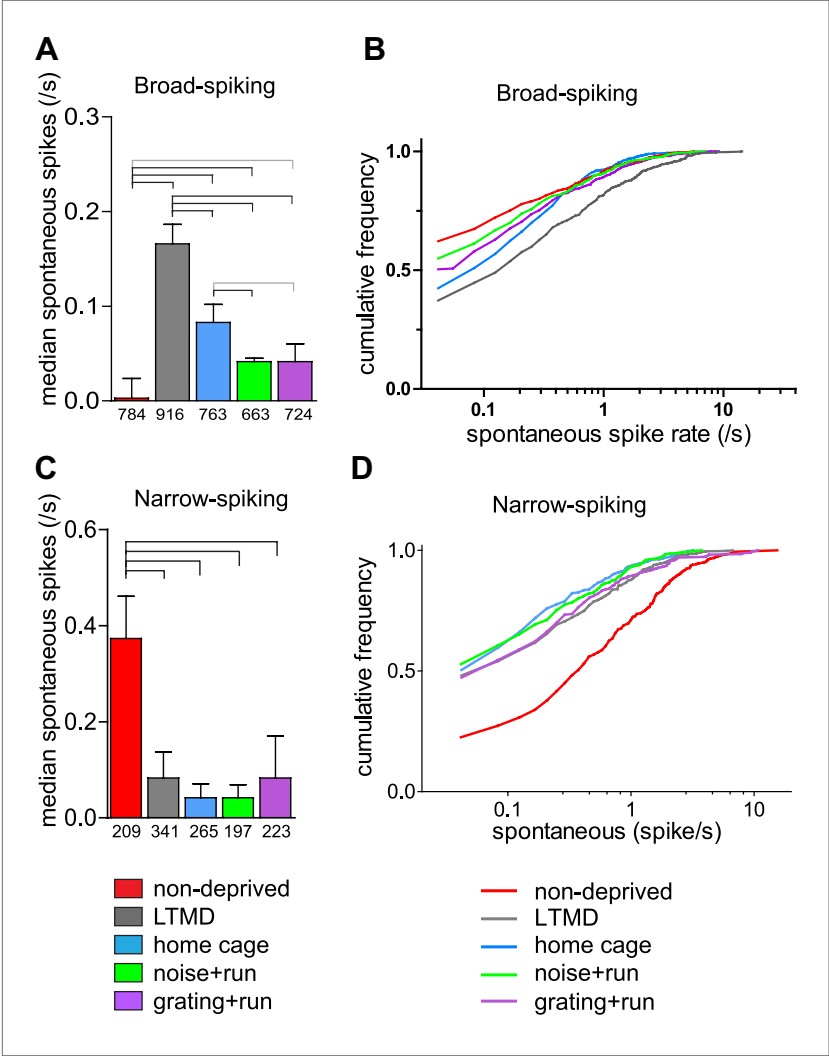

**Figure 7**. Spontaneous firing of isolated broad-spiking and narrow-spiking cells recorded simultaneously. (**A** and **B**) The median (±s.e.) number (**A**) and cumulative frequency distribution (**B**) of spontaneous firing rates of all broad-spiking cells isolated during the presentation of blank screen randomly interspaced in the drifting grating set. (**C** and **D**) The median (±s.e.) number (**C**) and cumulative frequency distribution of spontaneous firing rates of all narrow-spiking cells. Sample sizes are indicated in **A** and **C**. Horizontal lines above bars; black: p<0.01, grey: p<0.05. Results of Kolmogorov–Smirnov tests for **B** and **D** are shown in *Table 1*.

*2003*). Response amplitude was an average of at least four measurements. Ocular dominance index was computed as (R − L)/(R + L), where R and L are the peak response amplitudes through the right eye and the left eye, respectively, as described (*Kaneko et al., 2008*). All mice were kept under standard housing conditions with free access to food and water between recordings and daily running on the treadmill.

## Electrophysiological recording of single unit activities in vivo

Data acquisition, visual stimuli, and spike analyses were performed as described (*Niell and Stryker, 2008*) with minor modifications. Briefly, after locating the binocular area of the primary visual cortex by recording intrinsic signals elicited by ipsilateral eye stimulation, a small craniotomy (~2 mm diameter) was made over the binocular area through the window of the implanted headplate. A silicon multisite electrode with a tetrode configuration (model a2 × 2-tet-3mm-150-121, Neuronexus Technologies, MI) was inserted to a depth of <400 μm below the cortical surface to record cells in layer 2/3 and 4. Visual stimuli were generated in Matlab using the Psychophysics Toolbox extensions and displayed on a

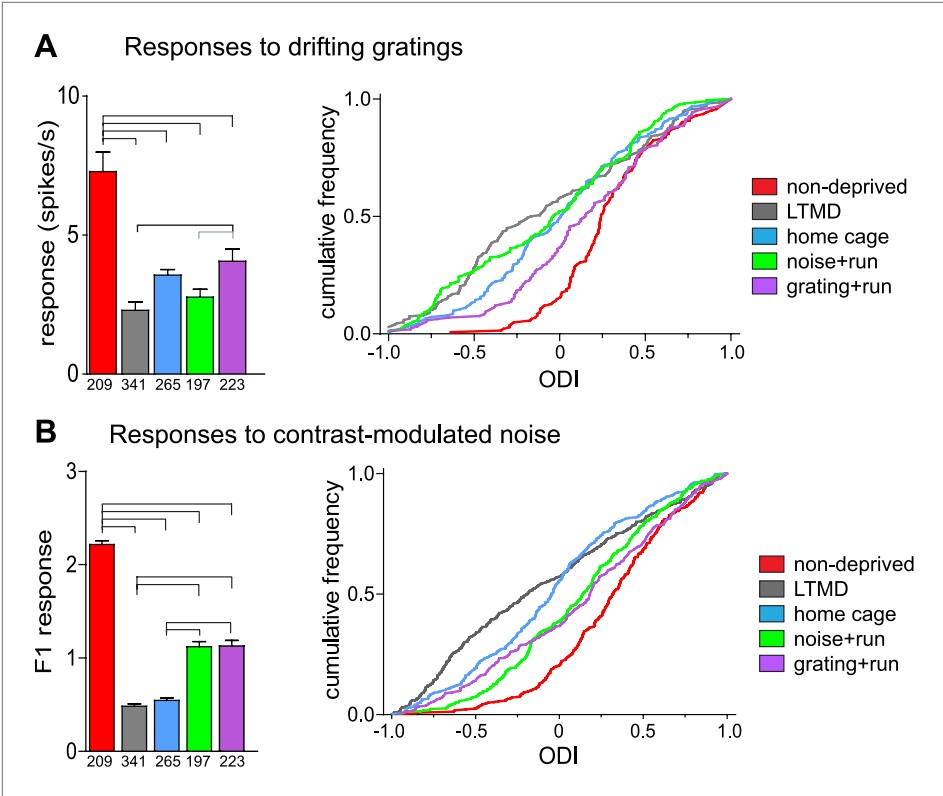

**Figure 8**. Responses of isolated narrow-spiking cells recorded simultaneously with broad-spiking cells. (**A**) Responses to drifting sinusoidal gratings of all narrow-spiking cells isolated. Left: median ±s.e. of peak responses to the optimal gratings through the deprived eye. Right: cumulative frequency distribution of ocular dominance index (ODI) calculated for each cell from peak responses through the deprived eye (as shown in **Figure 8—figure supplement 1A**) and the open eye (**Figure 8—figure supplement 1B**). (**B**) Responses to contrast-modulated noise in all narrow-spiking cells isolated. Left: median ±s.e. of F1 responses through the deprived eye. Right: cumulative frequency distribution of ocular dominance index (ODI) calculated for each cell from F1 responses through the deprived eye (as shown in **Figure 8—figure supplement 1C**) and the open eye (**Figure 8—figure supplement 1D**). Horizontal lines above bars; black: p<0.01, grey: p<0.05. Results of Kolmogorov–Smirnov tests for cumulative frequency distributions are shown in **Table 1**.

The following figure supplements are available for figure 8:

**Figure supplement 1**. Responses of isolated narrow-spiking cells recorded simultaneously with broad-spiking cells.

---

monitor placed 25 cm in front of the mouse as described above. In each animal in all treatment groups, the responsiveness of cells was measured using both drifting sinusoidal gratings and the contrast-modulated noise movie, as described above. Neuronal signals (spikes) were acquired using a System 3 workstation (Tucker–Davis Technologies, FL) and analyzed with custom software in Matlab (MathWorks, MA). Single units were identified by clustering spike waveforms using the FastICA Matlab package and KlustaKwik (**Harris et al., 2000**). Units were classified as broad- or narrow-spiking based on trough-to-peak time and the slope of the waveform 0.5 ms after the initial trough as described (**Niell and Stryker, 2008**).

Drifting sinusoidal gratings as described above (2.5 min/set) were repeated six times for each eye (30 min total). The average spontaneous firing rate for each unit was calculated by averaging the rate over all blank condition presentations. Responses to each orientation and spatial frequency were calculated by averaging the spike rate during the 1.5 s presentation and subtracting the spontaneous firing rate. For assessing ocular dominance, we first selected units responsive to the optimal grating (≥ 2.0 spikes/s) through either eye and then calculated ODI for each unit as the difference between the

contralateral (deprived) and ipsilateral (open) eyes divided by the sum of the two eyes' responses (*Kaneko et al., 2008*).

The preferred orientation was determined by averaging the response across all spatial frequencies, and calculating half the complex phase of the value $\dfrac{\sum F(\theta)e^{2i\theta}}{\sum F(\theta)}$. Given this fixed preferred orientation $\theta_{pref}$, the tuning curve was fitted with the sum of two Gaussians centered on $\theta_{pref}$ and $\theta_{pref} + \pi$, of different amplitudes $A_1$ and $A_2$ but equal width $\theta$, with a constant baseline B. From this fit, we derived an orientation selectivity index (OSI) and the width of the selectivity tuning. OSI was calculated as the depth of modulation from the preferred orientation to its orthogonal orientation $\theta_{ortho} = \theta_{pref} + \pi /2$, as $(R_{pref} - R_{ortho})/(R_{pref} + R_{ortho})$. Tuning width was the half-width at half-maximum of the fit above the baseline. The difference in preferred orientation between the two eyes was calculated by subtracting ipsilateral from contralateral preferred orientation along the 180° cycle (−90° ~ +90°). The absolute values of these differences were used in statistical analyses. The spatial frequency tuning curve was determined from the response at the preferred orientation ($\theta_{pref}$) described above and was fit to a difference of two Gaussians.

We probed the overall responsiveness of cells using movies of stochastic noise with defined spatial and temporal frequency spectra. The contrast of movies was periodically modulated so that each movie transitioned sinusoidally from a gray background to full contrast movie and back to gray again, with a 10-s period. This generally resulted in a periodic modulation of firing and also helped to maintain high firing rates throughout the presentation by preventing habituation. The movie was shown for 10–15 min for each eye, and the cells' responsiveness was presented as the response amplitude at the frequency of the contrast modulation (F1 response, the first harmonic at 0.1 Hz). A form of contrast–response curve was generated by plotting the averaged response over all cycles to the contrast. Average value of the contrast that elicits half-maximal response ($C_{1/2}$) was used to measure contrast sensitivity.

## Statistical analysis

Data were presented as mean ± SEM or median ± SE in the main text, as well as cumulative frequency distributions in the supplemental figures, unless otherwise indicated. Standard error of the median was calculated by a bootstrap. Response magnitudes and ocular dominance index data obtained from intrinsic signal imaging experiments were analyzed by using repeated measures ANOVA to assess changes from baseline as well as differences between treatments, which were followed by multiple comparisons with Bonferroni correction. Electrophysiological single unit data were analyzed using either one-way ANOVA (for normally distributed data) or Kruskal–Wallis one-way ANOVA (for the rest), followed by Bonferroni or Dunn multiple comparisons between treatments. Difference in cumulative frequency distribution was analyzed using the Kolmogorov–Smirnov test. These statistical analyses were performed using Prism 5 (GraphPad Software, CA) or Matlab (MathWorks, MA).

## Acknowledgements

This work was supported by NIH grant R01 EY02874 and T32 MH089920. We thank the members of the Stryker laboratory for critical reading, Cristopher Niell for assistance with microelectrode recording, and SW Zucker for suggesting the earth mover distance algorithm.

## Additional information

### Funding

| Funder | Grant reference number | Author |
| --- | --- | --- |
| National Institutes of Health (NIH) | R01 EY02874 | Michael P Stryker |
| National Institutes of Health (NIH) | T32 MH089920 | Megumi Kaneko |

The funders reviewed the study design as a specific aim of a grant application but had no role in data collection and interpretation or the decision to submit the work for publication.

## Author contributions

MK, Conception and design, Acquisition of data, Analysis and interpretation of data, Drafting or revising the article; MPS, Conception and design, Analysis and interpretation of data, Drafting or revising the article

## Ethics

Animal experimentation: This study was performed in strict accordance with the recommendations in the Guide for the Care and Use of Laboratory Animals of the National Institutes of Health. All of the animals were handled according to protocols approved by the UCSF institutional animal care and use committee under IACUC Protocol AN098080-01D.

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
