## [Decision Letter]

Thank you for sending your work entitled “Exercise promotes recovery of function in adult visual cortex” for consideration at *eLife*. Your article has been favorably evaluated by Eve Marder (Senior editor) and 2 reviewers, one of whom, Sacha Nelson, is a member of our Board of Reviewing Editors.

The Reviewing editor and the other reviewer discussed their comments before we reached this decision, and the Reviewing editor has assembled the following comments to help you prepare a revised submission.

The present study is a rigorous examination of the impact of locomotion and visual stimulation on the recovery of mice from amblyopia. The authors make the surprising and interesting (and potentially clinically important) discovery that pairing visual stimulation with the animal's own locomotion dramatically improves the responses through the eye that underwent long-term deprivation. Further, the authors go further to demonstrate that major forms of selectivity, including orientation and spatial frequency selectivity, are dramatically improved by the new protocol compared to locomotion or visual stimulation alone (or home-cage experience).

The reviews raised two major issues, both of which should be quite straightforward to deal with.

1) One feature of the results would benefit from clarification perhaps requiring additional analysis. The authors find that responses to a noise stimulus or bar stimulus are selectively enhanced by prior exposure to the same stimulus while running. They suggest that a Hebbian synaptic plasticity model may account for these results. This predicts that to the extent that the enhancement is stimulus specific, the stimuli should activate different circuits. One might imagine, for example, that the selective enhancement would be especially pronounced for cells that respond much better to one stimulus than the other, while cells that respond to both stimuli equally might be expected to show more balanced enhancement (although this need not be the case if the two stimuli activate the postsynaptic neuron through separate sets of synapses). Because the authors have recorded responses to both sets of stimuli in many individual cells they should be able to address this question of whether selective enhancement depends on selective response. This would potentially strengthen the interpretation in terms of an underlying plasticity mechanism. Although this analysis may be complicated by differences in the responses to gratings vs. bars, these differences are likely less prominent and presumably the authors have some handle on how these responses differ.

2) In the Title and Abstract, the authors refer to locomotion or running with the more generic term “exercise”. We appreciate that the authors are going for big picture appeal, but we are concerned that many readers will insert their own interpretation for “exercise”. In fact, the findings the authors report are related to a specific activity, locomotion/running for an extended period of time (several hours). In our opinion, the authors should use a specific term such as “running” or “locomotion”. The paper is very clear and careful; it would be shame for the conclusions to be misapplied by someone who only looks at the Title/Abstract. (Maybe “exercise” as a keyword?)

---

## [Author Response]

Thank you for the reviews of our manuscript. We have carried out the analysis suggested as major issue #1 and we have incorporated new text in the manuscript, a new paragraph in the Discussion, and a new supplemental figure (Figure 4—figure supplement 2) to address the conclusions that are sustained by this analysis. Following the suggestion outlined in the reviewers’ major issue #2, we have also revised the manuscript title and excised the word ‘exercise’ from our text at several points.